# Impact of Socioeconomic Inequalities on Dental Caries Status in Sardinian Children

**DOI:** 10.3390/children11010096

**Published:** 2024-01-12

**Authors:** Marco Dettori, Antonella Arghittu, Andrea Cappai, Paolo Castiglia, Guglielmo Campus

**Affiliations:** 1Department of Restorative, Pediatric and Preventive Dentistry, University of Bern, 3012 Bern, Switzerland; madettori@uniss.it; 2Department of Medicine, Surgery and Pharmacy, University of Sassari, 07100 Sassari, Italy; aarghittu@uniss.it (A.A.); paolo.castiglia@uniss.it (P.C.); 3Department of Architecture, Design and Urban Planning, University of Sassari, 07041 Alghero, Italy; a.cappai7@phd.uniss.it; 4Department of Cariology, Saveetha Dental College and Hospitals, SIMATS, Chennai 600077, India

**Keywords:** caries epidemiology, dmft, deprivation, spatial analysis, children

## Abstract

Background: The association between oral health of schoolchildren living in the North Sardinia area and socioeconomic deprivation was assessed to evaluate a potential spatial correlation. Methods: A total of 10,947 subjects were examined (5281 aged 3–5-years, and 5666 aged 6–11-years). The WHO dmft index score was calculated following clinical examination by calibrated examiners. The Sardinian Deprivation Index (IDMS) of the children’s municipalities was also considered. Descriptive, bivariate and multinomial data analysis was conducted to assess the association between clinical data and socioeconomic deprivation. The presence of systematic spatial variation regarding caries experience (dmft) and deprivation status was investigated using a spatial autoregressive analysis. Results: Caries figures were statistically different in the two age groups (dmf > 0, 13.79% in the younger group vs. dmf > 0, 34.20% in the older one, *p* < 0.01). In a multinomial logistic regression model for caries experience, all the covariates were statistically significantly associated (*p* < 0.01) in comparison with the base outcome “caries-free”. Linear regression analysis showed a dependence of dmft on IDMS (*p* < 0.01). Based on this equation, the dmft of the 39 municipalities that did not participate in the survey was estimated. IDMS was statistically significantly associated (*p* < 0.01) with caries prevalence in the spatial regression model. Conclusions: The deprivation index significantly increased the risk of caries for all categories of caries experience and prevalence compared to caries-free. The relationship between IDMS and caries data was also confirmed by spatial analysis.

## 1. Introduction

Dental caries result from a years-long chain of events in which clinical, microbiological, behavioral, and social factors play a role. In developed countries, over the past 40 years there has been a substantial decrease in the prevalence of dental caries in both children and adults [1,2]. Nevertheless, dental caries remains the most common oral disease and a major public oral health problem [3].

Several studies on health inequalities have shown that social deprivation and living environments play a major role in childhood dental caries [4,5,6,7].

In this regard, there is consensus across the literature that a child’s health status is affected by social determinants such as income, education, health care, the built environment (including housing), home neighborhood, community and family circumstances [8,9,10,11,12,13]. Health-related disadvantages, in fact, are transferred from one generation to the next, and the relationships between socioeconomic status and health can be observed from a very young age, with their effects manifesting even in the prenatal stage [14,15].

Moreover, parents’ limited awareness of the importance of oral hygiene, lack of knowledge about the transmission of oral pathogens, diet, and oral hygiene habits are also relevant in the etiology of dental caries in children, and are significant in the prevention of common oral diseases. It has been shown that infants whose mothers have poor oral hygiene and high levels of cariogenic bacteria have an increased risk of infection, and may therefore be more susceptible to developing caries in early childhood. Similarly, behavioral patterns assumed by parents, such as tooth brushing after meals, regular visits to the dentist, and a hypoglycemic diet, are factors positively associated with their children’s dental and oral health [10,14,15].

The provision of dental care for children differs greatly from country to country and within the same geographic area, and is largely absent in many developing countries and in regions defined as high deprivation index [16,17,18].

This association has been previously described in Italy, where several deprivation indicators have been developed as tools for a health planning framework by helping to describe the characteristics of the Italian population, with particular attention paid to the differences among local areas [10,11,12,13,19,20,21].

In this regard, in Italy’s territory there is a significant divide between regions in northern Italy and those in southern Italy. Sardinia, an island located in southern Italy, is currently seeing a sharp increase in the number of socially marginalized people; this is also attributable to the increase in absolute and relative poverty [22,23] partly exacerbated by the COVID-19 pandemic.

To assess socioeconomic differences at the area level, an Index of Multiple Deprivation was developed in the region (Indice di Deprivazione Multipla della Sardegna–IDMS) using data from the 2011 Sardinian General Population Census [22]. Taking elementary variables as a starting point, it was possible to adequately describe the characteristics of the Sardinian population, focusing on the differences between local situations. In fact, the index was constructed from socioeconomic variables, namely income disadvantage (absolute poverty), level of education, presence of primary services (e.g., schools, pharmacies, post offices, family doctors, etc.), unemployment rate, mortality, environmental pollution, and crime rate [22,23].

The IDMS has previously been used to assess the relationship, if any, between socioeconomic deprivation and influenza vaccination coverage in the over-65s for the specific urban and peri-urban area of the Municipality of Sassari, confirming the relationship between deprivation status and health outcomes (vaccination adherence) in the elderly population [24]. Given the distinctive features that define the island (elderly population, low birth rate, and low per capita income), placing it among the regions considered to be at high risk of socio-health inequalities, it is essential to follow a Health Equity Assessment process. Such a process would include investigating the possible relationship between deprivation and other health outcomes (e.g., dental caries) concerning at-risk population cohorts such as the childhood demographic target [25].

Based on these premises, the present study was designed as a cross-sectional observational survey of the oral health status of schoolchildren in northern Sardinia. In addition, the study aimed to describe the data on caries and its ecological association with socioeconomic deprivation and to assess a potential spatial correlation between the two. The method of measuring caries lesions chosen by the authors, in line with standardized diagnostic methods for the purpose of enabling comparison of caries status in different populations and/or countries around the world, was the dmft/DMFT (decayed (D), missing (M), and filled (F) teeth (T), where upper case denotes permanent dentition and lower case primary dentition), developed in 1938 [26] and adopted by the WHO with various advantages and disadvantages [27,28,29,30,31,32,33].

## 2. Materials and Methods

### 2.1. Study Setting

The study was carried out during the 2018/2019 school year (from September 2018 to June 2019) in Sardinia, Italy. Sardinia is the second-largest Mediterranean island, and the regional territory is divided into 5 provinces (Metropolitan City of Cagliari; Sassari; Nuoro; Oristano; South Sardinia) and 377 municipalities. The resident population during the survey period consisted of 1,611,621 individuals, the population target (3–11 years) consisted of 103,238 children, while in the study area, there were 35,076 children of which 17,028 (48.50%) were female.

Overall, 544 primary schools are present in the region and 157 in the study area [34]. The survey concerned the province of Sassari, which comprises the northern territory of the island covering an area of 7692 square kilometers (the largest province in Italy) (Figure 1). Ninety-two municipalities were involved (Appendix A).

### 2.2. Study Design and Sample Size

The present study was designed as a cross-sectional observational survey and, in line with Italian legislation, the study proposal was submitted to the Ethical Committee. The experimental protocols could start after 60 days, even in the absence of their reply, as it did not require ethical approval according to Italian law [35]. The local section of the Italian Medical Association sponsored the survey, which was designed by the University of Sassari in agreement with the Italian Dental Association.

Sample size was assessed using the freeware online application openepi (http://www.openepi.com Version 3 (accessed on 30 June 2018)), taking into account the Italian caries prevalence reported in the literature [10] with a population size equal to 35,076, a hypothesized frequency of 46.5%, and a confidence level of 97. The returned number was 8783 subjects, which was further increased by 10% to compensate for any unforeseen problems (i.e., excluding expected dropouts, incomplete or invalid records, students not present at the time of visits, etc.), for a total of 9961 subjects to be examined in the classroom. This strategy provided a sample that was self-weighting.

Each child’s parents/caregivers received a leaflet explaining the study’s aim and requesting the child’s participation. Only children with the consent form signed by parents/caregivers were examined.

In total, 35,076 children were recruited and a total of 10,947 were examined; children with no parental consent and not present in the classroom at the moment of the examination were excluded. The study sample thus represented 36.29% of the total study area population aged 3–11 years.

The survey was carried out after a theoretical calibration process with caries lesions detected in images. The calibration method involved two calibration levels: the first (L1) involved an inter-examiner agreement between four main investigators (MD, AA, PC, GC), the group leaders (GLs) at the following level; the second level (L2) involved four groups of 32 pediatric dentists and inter-examiner agreement assessment according to the GLs in each group. A total of 128 dentists acted as examiners. The strength of agreement associated with kappa statistics was labelled as described in the literature [36].

The survey was designed following the WHO methodology for oral health surveys [27]. Oral examinations were undertaken in school rooms using a dental mirror, a probe, and a headset light. The following characteristics of the primary dentition status were recorded: decayed (d), missing (m), and filled (f) teeth, and the dmft (d + m + f) index was then calculated for each subject.

The IDMS as deprivation index was used within a range from 0 (minimum value of deprivation) to 1 (maximum value of deprivation). This indicator had already been successfully used by the research team for similar purposes [24,34].

### 2.3. Statistical Analysis

Data were entered into Excel 2021 (Microsoft Office, Microsoft Corporation, Redmond, WS, USA) and analyzed using STATA^®^ 17.0 statistical software (StatCorp., Austin, TX, USA).

The variables IDMS and age were grouped into categories. The IDMS continuous variable was grouped into 4 categories of deprivation status: very low (0–0.14); low (0.15–0.23); medium (0.27–0.38); and high (0.39–1.00). Age was grouped into 2 categories: 3–5 year old subjects with only primary teeth, and 6–11 year old subjects with mixed dentition. Caries experience was converted as a dichotomous variable based on the dmft index (0 = caries-free; 1 = at least one tooth with a history of caries disease regardless of active lesion, tooth extracted for caries, or filled). Caries prevalence (d subgroup) was also transformed into a dichotomous variable (0 = not decayed; 1 = decayed). Caries experience was also coded as follows: caries-free subjects, subjects with an experience of 1–2 teeth, subjects with an experience of 3–4 teeth, and subjects with an experience of more than 4 teeth. Caries prevalence as regards d lesions was also coded with the same intervals [11,37,38].

Qualitative variables were described with absolute and relative frequencies. Associations between categorical variables were tested with Pearson’s chi-square. A nonparametric test for trend (linear-by-linear trend test) across exposure categories was calculated. Quantitative variables were represented by measures of position and variability. ANOVA one-way was used to evaluate the differences between parametric variables.

Multinomial (polytomous) logistic regression models were run to assess the relationships between dependent (caries experience and caries prevalence categories) and independent variables. For this analysis, the base outcome was “caries-free” subjects both for caries experience and prevalence.

The dependence between dmft and IDMS was evaluated by linear regression analysis. For municipalities with missing data, dmft was estimated through the same linear equation starting from the continuous figure of the variables.

All 92 municipalities were then inserted into a spatial autoregressive model, evaluating the spatial correlation between dmft and IDMS.

The statistically significant level was set at *p* < 0.05 for all the analyses.

### 2.4. Autoregressive Analysis

Spatial Autoregressive Models (SAR) were run using datasets that contain observations on geographical areas, and spatial autoregressive analysis was performed to describe the presence of systematic spatial variation in the study area regarding caries experience (dmft) and deprivation status.

The Sardinian map was retrieved from GeoportaleSardegna [39], and ArcGIS (version 10.8.2, Redlands, CA, USA) was used for geographic mapping and shape file elaboration. After that, the shape file was imported into STATA17.0^®^ statistical software and analyzed using the “spregress” command for spatial relationships.

## 3. Results

The strength of agreement, recorded as categories of kappa statistics, within raters and between raters and benchmarks was ‘substantial’ (0.81–0.90).

Overall, 54 municipalities out of 92 (58.70%) and 124 schools out of 157 (79.99%) in the area (see Figure 1) agreed to take part in the survey.

In total, 10,947 children were examined: 5281 in the younger group (3–5 years) and 5666 in the older group (6–11 years) (Table 1).

No statistically significant differences were observed regarding sex distribution between age groups (*p* = 0.65). Caries indicators were different in the two age groups, both for caries experience (dmf > 0 = 13.79% in the younger group vs. dmf > 0 = 34.20% in the older group, *p* < 0.01) and for caries prevalence (d > 0 = 12.65% in the younger group vs. d > 0 = 28.64% in the older group, *p* < 0.01) (Table 1). Caries scores, dmft, and subgroups were statistically significantly higher in the older group (Table 2). In particular, dmft = 1.07 ± 2.02 in the older group vs. dmft = 0.45 ± 1.55, *p* < 0.01 in the younger group.

The classification of the dmft and d subgroup were seen to be associated (Table 3) with sex, age groups, and deprivation (*p* < 0.01). Moreover, a linear trend across exposure categories was noted (*p* < 0.01).

For caries experience classification, the estimated coefficients of the multinomial (polytomous) logistic regression transformed to relative ratios were statistically significantly associated (*p* < 0.01) in comparison with the base outcome “caries-free” (Table 4).

Being female had a protective effect on caries prevalence classification. Very similar features were also present if caries prevalence classification (d) was used as a dependent variable; in subjects having more than four active caries lesions, the sex protective effect (being female) was not statistically significant (*p* = 0.07) (Table 4).

Linear regression analysis showed a dependence of dmft on IDMS (y = 1.1604x + 0.4679; *p* < 0.01).

Based on this equation, the dmft of the 39 municipalities that did not participate in the survey was estimated.

The autoregressive analysis showed the spatial relationship among the 92 municipalities (Figure 2).

The IDMS was statistically significantly associated (*p* < 0.01) with caries prevalence in the spatial regression model (Table 5).

## 4. Discussion

The role of socioeconomic status in the epidemiology of dental caries is quite noteworthy. Determinants such as educational level, lack of knowledge about oral health and eating habits, parents’ income levels, and limited access to dental services, influence the oral health status of children and adolescents [40,41]. Moreover, it is well known that children from socioeconomically deprived families show a higher prevalence of caries associated with more severe clinical manifestations than those from less deprived settings [10,42,43].

In Italy, few studies have focused on the specific health needs in specified administrative areas ripe for intervention. In Italy, caries in preschool children is statistically significantly linked to various socioeconomic indicators (e.g., GNI, GINI, unemployment rate) [10].

This study focuses at the level of municipalities using a purpose-built indicator [22].

This study aimed to investigate the experience and prevalence of primary tooth caries (dmft) among preschool (3–5 years) and school-age (6–11 years) children residing in northern Sardinia by describing the geographical association of caries data with socioeconomic indicators. A secondary objective was to assess the potential spatial autoregressive pattern between the recorded dmft and the Sardinian deprivation index in the micro areas of interest (municipalities in the province of Sassari).

The study area (Sardinia), due to the insularity and territorial contiguity which preserves the region from outside interferences, is an optimal setting for investigating social and epidemiological dynamics [44].

In the past, studies on single towns in Sardinia have shown a link between caries risk and socioeconomic factors, although geographical variations outside urban areas were never assessed. Thus, there was a need for a new comprehensive cross-sectional epidemiological study to understand the current trend of caries in Sardinian children.

The sample enrolled in the present survey was homogeneous regarding gender distribution among the age range. Caries experience (dmft) was statistically significantly higher in children aged 6–11 years than in younger ones as expected, as caries is a developmental and cumulative event; a greater experience of caries tends to be observed as the years progress [45].

The caries data from this survey underline that in the study area, the prevalence of caries is higher than in other Italian regions.

The largest contribution to the dmft index comes from the ‘decayed’ component (d), underlining the continuing need for intervention within the study area, even though studies show a general improvement in the population’s oral health conditions over time [41].

The sub-category analysis of caries experience showed a statistically significant as-sociation with sex (with a higher prevalence of caries-free subjects being female) and age (with dmft values ≥1 almost trebling in school-age subjects). In older subjects, the need for treatment is more than twice as high as it is in younger subjects. The protective effect of sex (being female) is more evident in the logistic regression analysis, except with active caries values above 4, in which case the effect does not reach significance. This can be attributed to the low number of females with high caries lesions.

To the authors’ knowledge, this is the first study to consider the role of a precise deprivation index (IDMS) in the epidemiological study of dental caries in several municipalities in Sardinia. Previous analyses have indicated that socioeconomically disadvantaged individuals are greatly affected by oral health problems, and determinants such as low socioeconomic status, minority status, and unemployment, are associated with low levels of preventive dental care and high rates of dental disease [4,41,45,46]. The findings of this paper emphasize that deprivation scores are significantly associated with caries [11,27].

Furthermore, logistic regression analysis revealed that IDMS significantly increased the risk for all categories of caries experience and prevalence compared to caries-free subjects. From this perspective, deprivation indices are useful to identify small areas with high levels of need for dental care and increased oral health promotion and prevention services [12,21].

The spatial relationship between caries experience and IDMS returned a statistically significant association, as mentioned above. Spatial autocorrelation analysis enabled observation of the phenomenon’s distribution throughout the entire study area, albeit in the absence of a clustering effect [46].

The study’s main limitations can be ascribed to the typical characteristics of an observational study. The study design is not suitable for testing etiological hypotheses but only for formulating them, as it does not allow an assessment of the cause-effect relationship (i.e., whether the supposed factor causes the disease or vice versa). The present survey aimed to (i) describe the distribution of caries at a population level, and (ii) to associate the data with certain risk factors. The huge number of subjects enrolled and examined enabled us to infer the outcomes for the whole population.

The deprivation values were significantly associated with caries data. In particular, logistic regression analysis showed that the deprivation index significantly increased the risk for all categories of caries experience and prevalence compared to caries-free subjects. The relationship between IDMS and caries data was further confirmed by spatial analysis.

This study again adds emphasis to the pressing need to push for educational interventions, particularly in schools where, due to compulsory education, the population is more easily reached. In addition, the study has made it possible to detail those areas in North Sardinia in greatest need of intervention, and this can make equity-based primary prevention strategies more efficient.

## Figures and Tables

**Figure 1 children-11-00096-f001:**
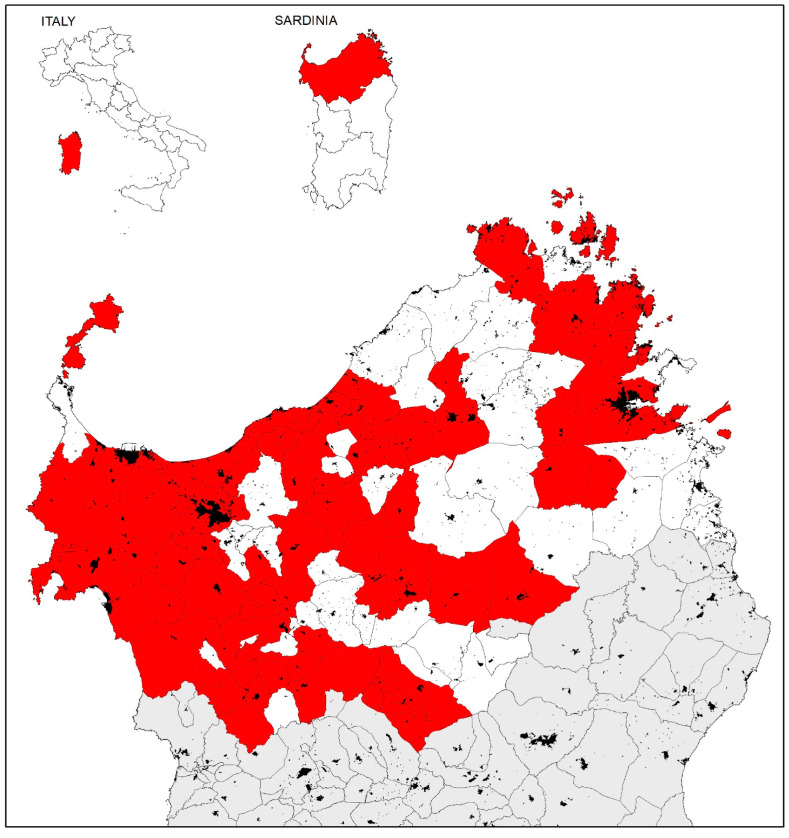
Map of Italy, Sardinia, and municipalities involved in the study.

**Figure 2 children-11-00096-f002:**
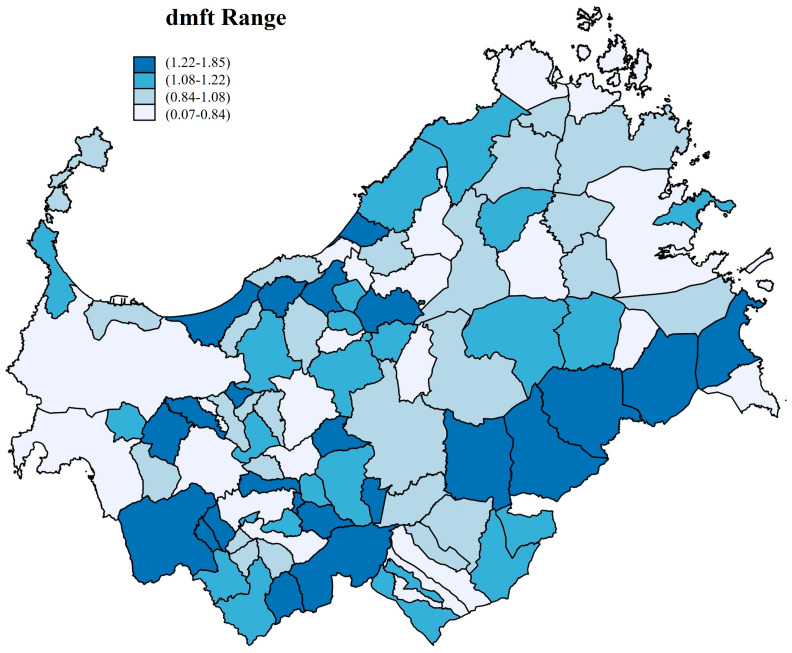
Spatial analysis among the 92 municipalities by dmft range.

**Table 1 children-11-00096-t001:** Sample characteristics according to age groups (3–5 years and 6–11 years) sorted by sex, caries experience (dmf > 0), and prevalence (d > 0).

Pearson chi-square was used to evaluate the association between groups.			
	3–5 years *n* (%)	6–11 years *n* (%)	Total *n* (%)
	Sex		
Males	2677 (50.69)	2897 (51.13)	5574 (50.92)
Females	2604 (49.31)	2769 (48.87)	5373 (49.08)
	Pearson *χ*^2^_(1)_ = 0.21 *p* = 0.65		
	Caries Experience		
Caries-free (dmf = 0)	4553 (86.21)	3728 (65.80)	8281 (75.65)
Affected (dmf > 0)	728 (13.79)	1938 (34.20)	2666 (24.35)
	Pearson *χ*^2^_(1)_ = 618.59 *p* < 0.01		
	Caries Prevalence		
d = 0	4613 (87.35)	4043 (71.36)	8656 (79.07)
d > 0	668 (12.65)	1623 (28.64)	2291 (20.93)
	Pearson *χ*^2^_(1)_ = 422.61 *p* < 0.01		

**Table 2 children-11-00096-t002:** Caries index dmf and subgroups (means and Standard deviation) in the two age groups.

	3–5 Years Mean ± SD	6–11 Years Mean ± SD	ANOVA One Way F-Value *p*-Value
decayed (d)	0.41 ± 1.48	0.81 ± 1.76	F = 163.40 *p* < 0.01
missing (m)	0.01 ± 0.11	0.10 ± 0.35	F = 89.49 *p* < 0.01
filled (f)	0.03 ± 0.31	0.20 ± 0.77	F = 222.96 *p* < 0.01
dmf	0.45 ± 1.55	1.07 ± 2.02	F = 317.42 *p* < 0.01

**Table 3 children-11-00096-t003:** Association among caries experience (dmf) and prevalence (d) classifications and sex, age groups, and deprivation index.

-A Caries Experience					
	dmf = 0 *n* (%)	dmf = 1–2 *n* (%)	dmf = 3–4 *n* (%)	dmf > 4 *n* (%)	Total
	Sex				
Males	4105 (73.65)	748 (13.42)	407 (7.30)	314 (5.63)	5574 (50.92)
Females	4176 (77.72)	648 (12.06)	297 (5.53)	252 (4.69)	5373 (49.08)
	Pearson *χ*^2^_(3)_ = 28.07 *p* < 0.01 Linear trend *z* = −4.90 *p* < 0.01				
	Age groups				
3–5 years	4553 (86.21)	400 (7.57)	162 (3.07)	166 (3.14)	5281 (48.24)
6–11 years	3728 (65.80)	996 (17.58)	542 (9.57)	400 (7.06)	5666 (51.76)
	Pearson *χ*^2^_(3)_ = 625.73 *p <* 0.01 Linear trend *z* = 22.01 *p* < 0.01				
	Deprivation Index *				
Very low	2885 (80.90)	385 (10.80)	157 (4.40)	139 (3.90)	3566 (32.58)
Low	3216 (79.88)	454 (11.28)	189 (4.69)	167 (4.15)	4026 (36.38)
Medium	560 (61.07)	168 (18.32)	119 (12.98)	70 (7.63)	917 (8.37)
High	1620 (66.45)	389 (15.96)	239 (9.80)	190 (7.79)	2438 (22.27)
	Pearson *χ*^2^_(3)_ = 343.49 *p <* 0.01 Linear trend *z* = −4.04 *p < 0*.01				
-B Caries Prevalence					
	d = 0 *n* (%)	d = 1–2 *n* (%)	d = 3–4 *n* (%)	d > 4 *n* (%)	Total
	Sex				
Males	4317 (77.45)	706 (12.67)	323 (5.79)	228 (4.09)	5574 (50.92)
Females	4339 (80.76)	609 (11.33)	231 (4.30)	194 (3.61)	5373 (49.08)
	Pearson *χ*^2^_(3)_ = 21.54 *p* < 0.01 Linear trend *z* = 13.73 *p <* 0.01				
	Age groups				
3–5 years	4613 (87.35)	369 (6.99)	148 (2.80)	151 (2.86)	5281 (48.24)
6–11 years	4043 (71.36)	946 (16.70)	406 (7.17)	271 (4.78)	5666 (51.76)
	Pearson *χ*^2^_(3)_ = 431.98 *p* < 0.01 Linear trend *z* = 16.97 *p <* 0.01				
	Deprivation Index *				
Very low	2977 (83.48)	365 (10.24)	115 (3.22)	109 (3.06)	3566 (32.58)
Low	3346 (83.11)	411 (10.21)	144 (3.58)	125 (3.10)	4026 (36.38)
Medium	600 (65.43)	176 (19.19)	88 (9.60)	53 (5.78)	917 (8.37)
High	1733 (71.08)	363 (14.89)	207 (8.49)	135 (5.54)	2438 (22.27)
	Pearson *χ*^2^_(3)_ = 308.47 *p* < 0.01 Linear trend *z* = 13.73 *p* < 0.01				

* Deprivation Index was categorized as follows: index 0–0.14 very low deprivation; index 0.15–0.23 low deprivation; index 0.27–0.38 medium deprivation; index 0.39–1.00 high deprivation.

**Table 4 children-11-00096-t004:** Multinomial ordinal logistic regression model using caries experience (dmf) and prevalence (d) as dependent variables and classifications and sex, age groups, and deprivation index.

	RRR (SE)	*p*-Value	_95%_CI
-A Caries Experience N obs = 10,947 Log Likelihood = −8392.96 *χ*^2^(9) = 803.40 *p* < 0.01			
dmf = 0	Base outcome		
dmf = 1–2			
Sex (females)	0.84 (0.05)	<0.01	0.74–0.95
Age groups (6–11 yy)	2.86 (0.18)	<0.01	2.52–3.24
Deprivation index (high)	1.16 (0.03)	<0.01	1.10–1.22
Constant	0.03 (0.01)	<0.01	0.02–0.04
dmf = 3–4			
Sex (females)	0.70 (0.05)	<0.01	0.60–0.82
Age groups (6–11 yy)	3.60 (0.34)	<0.01	3.00–4.33
Deprivation index (high)	1.36 (0.05)	<0.01	1.27–1.45
Constant	0.01 (0.01)	<0.01	0.01–0.01
dmf > 4			
Sex (females)	0.77 (0.07)	<0.01	0.65–0.92
Age groups (6–11 yy)	2.63 (0.25)	<0.01	2.18–3.18
Deprivation index (high)	1.30 (0.05)	<0.01	1.21–1.41
Constant	0.01 (0.01)	<0.01	0.01–0.02
-B Caries Prevalence *N* obs = 10,947 Log Likelihood = −7549.30 *χ*^2^_(9)_ = 593.84 *p <* 0.01			
	RRR (SE)	*p*-value	95%CI
d = 0	Base outcome		
d = 1–2			
Sex (females)	0.85 (0.05)	<0.01	0.76–0.96
Age groups (6–11 yy)	2.75 (0.18)	<0.01	2.42–3.13
Deprivation index (high)	1.15 (0.03)	<0.01	1.09–1.22
Constant	0.03 (0.01)	<0.01	0.02–0.04
d = 3–4			
Sex (females)	0.70 (0.06)	<0.01	0.85–0.83
Age groups (6–11 yy)	3.60 (0.34)	<0.01	2.20–2.26
Deprivation index (high)	1.44 (0.06)	<0.01	1.33–1.56
Constant	0.01 (0.01)	<0.01	0.01–0.01
d > 4			
Sex (females)	0.84 (0.08)	0.07	0.68–1.02
Age groups (6–11 yy)	1.84 (0.19)	<0.01	1.50–2.26
Deprivation index (high)	1.28 (0.06)	<0.01	1.17–1.40
Constant	0.01 (0.01)	<0.01	0.01–0.02

**Table 5 children-11-00096-t005:** Spatial regression model of caries experience (dmf) and deprivation index (IDMS).

N obs = 92 municipalities Wald *χ*^2^_(1)_ = 6.93 *p* < 0.01			
Caries prevalence	Coef (SE)	*p*-value	_95%_CI
IDMS	0.32 (0.12)	<0.01	0.08–0.55
Constant	0.96	<0.01	0.88–1.05

## Data Availability

The data are available on request from the corresponding author due to privacy restrictions.

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
