# Peer review of "Impact of Socioeconomic Inequalities on Dental Caries Status in Sardinian Children"

_children, 2024, doi:10.3390/children11010096_

Round 1

Reviewer 1 Report

Comments and Suggestions for Authors

Well done to the authors for an excellent study.

My only comment would be to provide some context, and how the findings of the study can affect Sardinian children. Furthermore, the authors should highlight the clinical significance and impact of their findings. 

Reviewer 2 Report

Comments and Suggestions for Authors

Dear authors, 

Manuscript title: "Impact of Socio-Economic Inequalities on Dental Caries Status in Sardinian Children" needs modification. 

Abstract: well-written and presented. Keywords: understandable and within SEO.

Introduction: Too short paragraphs. Lines 40–42 do not make any sense; rewrite it. The introduction does not clarify the need for the study. The authors have not explained how it is going to impact the caries index in children with low socioeconomic status. As it is known from the literature, children with low socioeconomic status have a high caries index. So what does the author want to present here?

Material and method: The authors should present "IDMS as a deprivation index." who filled out these forms? Provide this form as data.

Result: Well-presented. 

Discussion: Refer to the studies that have utilized the IDMS index and how they were helpful. The clinical relevance of this finding. How will it be helpful to policymakers? What steps should a municipality take to reduce the caries burden?

The conclusion is not presented clearly.

More references should be added and the importance of IDMS should be explained clearly. 

Comments on the Quality of English Language

Minor editing is required. 

Reviewer 3 Report

Comments and Suggestions for Authors

Dear Authors, this paper about the impact  of socio-economic inequalities on dental caries status in Sardinian children is really interesting and well performed. I am sure that both scientists and professionals will find it helpful for their purposes.

Some issues need to be solved before its final publication in the journal.

Introduction:  This is a really important part of an article and it should enraged in your paper. I suggest to add some paragraphs about how parents knowledge of dental health can influence their children oral health. This paper can be used: Ludovichetti FS, Zuccon A, Lucchi P, Cattaruzza G, Zerman N, Stellini E, Mazzoleni S. Mothers' Awareness of the Correlation between Their Own and Their Children's Oral Health. Int J Environ Res Public Health. 2022 Nov 14;19(22):14967.

Chalvatzoglou E, Anagnostou F, Arapostathis K, Boka V, Arhakis A. Assessment of Young Mothers' Oral Hygiene Practices during Pregnancy and Their Knowledge of Children's Oral Health in Northern Greece. J Contemp Dent Pract. 2023 Mar 1;24(3):202-206.

Materials and methods and results are well written and easy to understand

 Improve clarity by providing more context on socioeconomic indicators and their collective influence on dental caries in children. Emphasize the study's unique contribution in investigating geographical variations within Sardinia, highlighting the need for a comprehensive approach.

 Enhance the discussion by delving into the significance of observed higher caries prevalence in the study area, explaining the dmft index's implications, and discussing the ongoing need for treatment despite oral health improvements.

Strengthen the discussion by thoroughly exploring study limitations, acknowledging the observational design's constraints, addressing potential confounding factors, and suggesting future research directions. Emphasize practical implications, such as using the deprivation index to target areas for dental care intervention and prevention.

Round 2

Reviewer 2 Report

Comments and Suggestions for Authors

All the issues have been addressed.